# Shipping noise tolerance in invertebrates: A case study of the shore crab *Hemigrapsus oregonensis*

Abigail Birch[1], Kieran D. Cox[2,3], Kelsie A. Murchy[3], Sandra Emry[1], Christopher D. G. Harley[1,4]*

1 Department of Zoology, University of British Columbia, Vancouver, British Columbia, Canada,
2 Department of Biological Sciences, Simon Fraser University, Burnaby, British Columbia, Canada,
3 Department of Biology, University of Victoria, Victoria, British Columbia, Canada, 4 Institute for the Oceans and Fisheries, University of British Columbia, Vancouver, British Columbia, Canada

* harley@zoology.ubc.ca

## Abstract

Recent decades have seen significant alterations to ocean soundscapes. These changes are primarily driven by human-generated sources (i.e., anthropogenic noise), which is now recognized as a marine pollutant of emerging concern. In contrast to research on marine mammals and fish, studies on the effects of noise on marine invertebrates are limited, and while behavioural changes have been observed in some invertebrate taxa, few investigations have considered whether marine invertebrates can develop a tolerance to this pollutant. We examined the behavioral impacts of shipping noise on the shore crab *Hemigrapsus oregonensis* and whether they can develop tolerance to noise. Cohorts collected from sites with low and high noise levels were exposed to playbacks of ship noise in the laboratory. We measured initial responses to a simulated predator attack, time taken to seek shelter following the attack, and disruption during feeding. Our results indicated that ship noise significantly impacts shore crabs' initial response after a simulated predator attack, with a 66% likelihood of movement in noise-exposed individuals compared to 32% in the controls. However, ship noise did not significantly impact whether the crabs retreated to shelter after a predator attack, nor did it disrupt feeding. The interaction between treatment and site type was not significant for any of the behavioral metrics, indicating no evidence of tolerance related to prior noise exposure. Finally, we assessed broader relationships between sound and marine arthropods' behavior by combining our results with 71 data points extracted from 17 published studies. A meta-analysis of these data indicated that sound can have a positive, negative, or null effect on marine arthropods. Our results highlight the importance of considering marine invertebrates when evaluating the ecological impacts of anthropogenic noise, and suggest that more work is required to identify the contexts in which this emerging pollutant is particularly detrimental.

**Data availability statement:** The data has been shared to the public repository Figshare. It is available through the following doi: http://doi.org/10.6084/m9.figshare.29090081.

**Funding:** Funding was provided by an NSERC Discovery Grant (RGPIN-2022-04683) to CDGH and the Liber Ero Fellowship Program to KDC. AB was supported by an Ecosystems and Oceans Science Contribution Framework grant awarded to KDC and Francis Juanes.

**Competing interests:** The authors have declared that no competing interests exist.

## Introduction

The last century was an era characterized by heightened globalization and industrialization, with global exports increasing 40-fold since 1913 [1]. As a result, commercial shipping has emerged as a ubiquitous feature of the world's oceans [2,3]. As of 2021, 90% of global exports were transported by sea, with shipping rates projected to continue to increase in the coming decade [4]. In the Northeast Pacific Ocean, low-frequency sound levels have increased by 3dB per decade since the 1960s, primarily attributed to the surge in commercial shipping traffic [5]. As a result, the marine soundscape has been significantly altered, impacting the organisms that occupy these ecosystems [6–8]. Disruptions to the soundscape and impacts on marine species have led to the recognition that shipping noise is a pervasive marine pollutant [9]. While research has primarily focused on the impact of elevated noise levels on acoustic cue-reliant organisms, particularly marine mammals [10], there has been increasing concern about how anthropogenic noise affects marine invertebrates [8–9]. However, given the diversity of invertebrate taxa, further investigation is imperative to understand the impact of noise pollution on these species, and, consequently, the health of marine ecosystems.

One such ecologically and economically important group of invertebrates is the crustaceans. Despite our limited understanding of crustaceans' auditory perception, recent evidence suggests that sound plays a vital role in their behavior and survival [11,12]. Research indicates that several organs in crustaceans, such as setae on the body surface, chordotonal organs, and statocysts within the antennules, are all capable of detecting particle motion caused by sound vibrations [13]. Additionally, antennae are believed to be involved in sound perception in crustaceans [14]. Unlike other aquatic animals, crustaceans do not have gas-filled cavities and are, therefore, less sensitive to the pressure component of sound. Crustaceans are likely most responsive to lower frequency sounds; however, significant variation exists among species [15]. For example, fiddler crabs (*Uca spp.*) are most sensitive to frequencies between 300 Hz-700 Hz [16], ghost crabs (*Ocypode spp.*) are most sensitive to frequencies between 1,000–2,000 Hz [17], and the Norway Lobster (*Nephrops noruegicus*) had highest sensitivities between 20–200 Hz [18].

Previous research investigating the effects of noise on green crabs (*Carcinus maenas*, also known as shore crabs in Europe) has demonstrated that it has varying effects on their behavior and physiology [10,19–22]. Shipping noise (148–155 dB re 1 µPa) can increase oxygen consumption, indicating higher metabolic rates and potentially elevated stress levels [19]. Anthropogenic vibrations have also been shown to increase antennae beat activity in green crabs, an indicator of stress due to the sensitivity of antennae to sound vibrations [20]. Sex-specific responses to anthropogenic vibrations have been demonstrated, with males exhibiting more activity than females, suggesting that they may experience greater stress in the presence of noise [20]. Furthermore, shipping noise has been shown to affect the sheltering response of both adult [21] and juvenile [10] green crabs, resulting in an increased time to retreat to shelter after a simulated predator attack. Additionally, for juvenile green crabs,

shipping noise reduced the likelihood of responding to a simulated predator attack [10], which was not seen in adults [21]. Noise has also been found to cause crabs to suspend feeding activities, however, foraging success and duration were not disrupted [21,22].

While anthropogenic noise has been shown to influence crabs negatively, these impacts can be mitigated if individuals can develop tolerance, allowing them to endure its presence with reduced adverse effects [23]. Tolerance is recognized as a moderation in response to a disturbance, manifesting as a behavioral state that can be measured at a single point in time [23]. Tolerance to noise pollution has been observed in different marine mammals, depending on the species and their previous noise exposure [24,25]. Similarly, tolerance to noise pollution has also been demonstrated in various marine fish species, including coral reef fish [26] and seabass [27]. However, research on noise tolerance in marine invertebrates, particularly crustaceans, is limited and warrants investigation. With shipping noise pervasive in marine environments, organisms likely endure chronic exposure, particularly in high traffic areas, or at least encounter intermittent exposure in quieter regions. Consequently, processes such as tolerance may be underway [23]. Wale et al. [19] tentatively supported the hypothesis of tolerance to shipping noise in green crabs (*Carcinus maenas*), as there was a slight decrease in oxygen consumption (a proxy for stress response) with each subsequent exposure to ship noise. However, the crabs were exposed to shipping noise eight times, making it challenging to determine whether the observed behavior signifies tolerance [19]. Rather than assessing tolerance solely through repeated lab-based playbacks, an alternative approach involves conducting a comparative analysis of individuals from high-noise and low-noise areas. This method may provide more substantial insights into tolerance levels by addressing the lack of prolonged exposure in lab-based tolerance studies [23].

Here, we investigate whether the shore crab *Hemigrapsus oregonensis* can develop a tolerance to shipping noise by examining individuals from both low (105.6–118.5 dB re 1 µPa) and high noise (123.4–134.2 dB re 1 µPa) sites. Tolerance was defined as significantly reduced responses to noise in comparison to controls. We used three metrics to assess tolerance to noise: initial response to a simulated predator attack, time taken to seek shelter following the attack, and disruption during feeding. We hypothesized that noise would increase the time to retreat to shelter after a predator attack and cause disruption during feeding. Additionally, we hypothesized that crabs collected from high noise sites would exhibit increased tolerance levels compared to crabs from low noise sites due to the differences between these populations' previous noise exposure. We then evaluated our findings relative to the published literature by conducting a meta-analysis of data extracted from 18 studies (including ours) on how sound influences marine arthropods' behavior. This research provides insight into the consequences of noise pollution on shore crabs, thereby enhancing our understanding of its broader impacts on benthic marine ecosystems.

## Methods

### Site selection

Nine sites were initially surveyed from rocky intertidal shores in Burrard Inlet, Howe Sound, and the southern Strait of Georgia, British Columbia, as *H. oregonensis* are ubiquitous across these habitats. Six of these sites (Barnet Marine Park (BMP), Girl In A Wetsuit (GIW) Beach, Horseshoe Bay Ferry Terminal, Sunset Beach, Tsawwassen Ferry Terminal and Whytecliff Park) were expected to be high noise sites, as they are located in narrows at the entrances to active port and marina facilities. The remaining three sites were expected to be low noise sites, either being located in lower traffic embayments (Brunswick Beach, Crescent Beach) or separated from shipping channels by a ~1 km wide intertidal sand bank (Acadia Beach). To confirm that noise levels matched our expectations based on proximity to vessel traffic, each site was visited once, either in October or November 2023. At each site, a one-hour recording was made between 09:30 and 17:00 using an underwater hydrophone (Cetacean Research Technology SQ-26; sensitivity: −169 dB re 1 V/µPa; flat frequency response (± 1 dB) up to 28 kHz, frequency range: 0.020–50 kHz) connected to a ZOOM recorder (H1 Handy Recorder, sampling rate = 48 kHz, 16-bit) at one meter depth. The recordings were imported into Audacity 3.3.3, and a relative sound pressure level (SPL) was calculated for each site. The SPL values were calculated in 15-minute intervals

by subtracting the hydrophone sensitivity (−169 dB re 1V/µPa), from Root Mean Square (RMS) dB, obtained from Audacity. Six of the nine sites were then chosen to be included in the study, based on their classification as either the loudest or quietest locations among those sampled (Fig 1A), coupled with the presence of a suitable population of *H. oregonensis* for collection (Fig 1A, Supplementary Material S1 Table in S1 File). Acadia Beach, Brunswick Beach, and Crescent Beach were confirmed as low noise sites (< 120 dB re 1 µPa), whereas Barnet Marine Park (BMP), Girl In A Wetsuit (GIW) Beach, and Sunset Beach were confirmed as high noise sites (> 120 dB re 1 µPa; Fig 1B). To ensure consistency of other environmental parameters across sites, salinity and temperature measurements were taken at all chosen locations using a thermometer and handheld refractometer (PCE–0100). Salinity levels and temperatures ranged from 24–25 psu and 7–9°C, respectively (refer to Supplementary Material S2 Table in S1 File).

## Noise treatments

Sound recordings of ships transiting by each collection site were used to create experimental noise tracks in Audacity 3.3.3. A six-minute track was made using three different ship types, which included a recreational fishing vessel, a tanker, and a ferry. Two different versions of the audio track were used during the experiment, with the order of the ships differing in each track. Ambient and playback sound levels in the tank were measured before the experiments began using an underwater hydrophone (SQ-26) connected to a ZOOM recorder (H1). Tank relative SPLs were calculated using the method described in Site Selection, and Power Spectral Density (PSD) analysis of sound playback was conducted in Python (version 3.7). $SPL_{RMS}$ in the tank ranged from 149–152 dB re 1 µPa for the noise playback and 120 dB re 1 µPa for ambient noise (Fig 2C, 2D, 2E). Based on previous research on hearing ranges in different crustacean species, the noise track likely contained frequencies that the shore crabs heard [15]. However, only sound pressure levels were measured, limiting insights into the particle motion aspect of the experiments, which crabs are most sensitive to [28].

## Experimental design

*Hemigrapsus oregonensis* were collected from the six sites between the 5th and 7th of January 2024 (scientific collecting license XMCFR 36 2023). A total of 32 crabs were collected from each site. Crabs were transported to the University

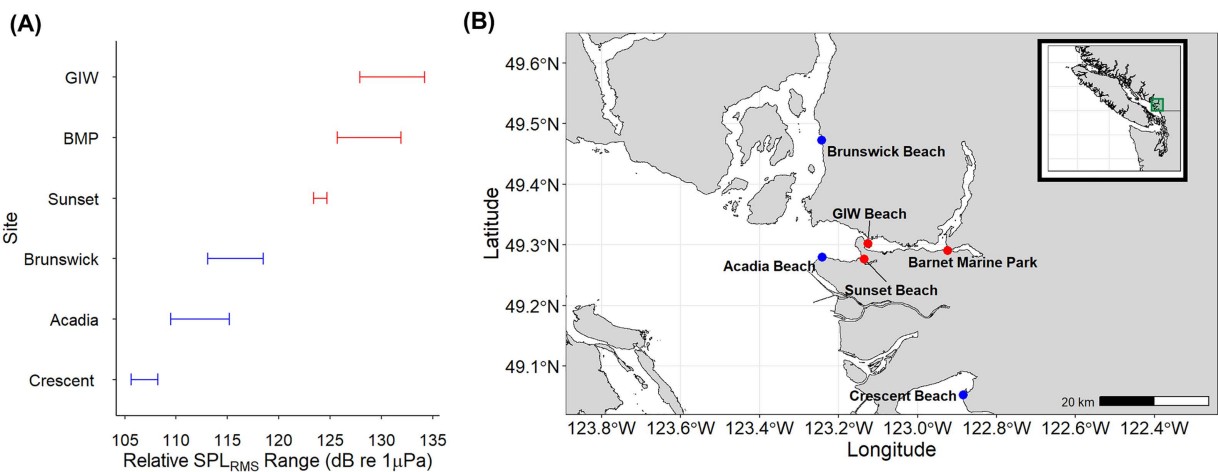

**Fig 1. Acoustic analysis of study sites in the Lower Mainland, BC. (A) Relative sound pressure levels at selected sites based on one-hour underwater recordings made between 09:30 and 17:00 using a SQ-26 hydrophone and ZOOM recorder.** Relative SPL was calculated in 15-minute intervals by subtracting Root Mean Square (RMS) dB values from hydrophone sensitivity (dB re 1V/uPa). **(B)** Geographic distribution of the six sites in the Lower Mainland, BC. Low noise sites are marked with blue pins; high noise sites are marked with red pins. Map images were generated using publicly available data from the British Columbia Marine Conservation Analysis Team's Marine Atlas of Pacific Canada (available via www.bcmca.ca).

of British Columbia in polystyrene containers (30 x 30 x 13 cm) to reduce excessive noise exposure and were covered in cool, damp paper towels. Holding tanks in the lab (45 x 25 x 30 cm) were wrapped with polystyrene blocks to reduce noise transmission. Tanks were filled with seawater kept at 12°C and salinity 33 psu to mimic natural conditions and minimize stress. Tank openings were covered to prevent crabs from escaping, and seawater was aerated using an air stone suspended from the cover. Water changes were done once per week. Crabs were fed~25g Thrive Hermit Crab Pellet Diet once per week. Crabs were acclimated in the holding tanks for one week before experimental trials began. All crabs were returned to their original collection sites following the conclusion of the experiment.

The experimental tank (77 x 32 x 54 cm) had an AQ339 Diluvio™ Underwater Loudspeaker placed 37 cm from the end wall at 26 cm depth (Fig 2A, 2B). A mesh wall was placed in the middle of the tank, creating an experimental arena on one side (38 x 32 x 54 cm). The tank was filled to approximately 32 cm depth with seawater and was cooled to 12–13°C using Ziploc bags of ice. The bottom of the tank was covered with aquarium gravel to provide substrate for the crabs. To minimize stress from human presence, a barrier was placed on the front of the tank to shield the observer from view. For experimental trials, a single crab was moved to the experimental tank, and the trial started immediately. Each crab was randomly allocated to one of the two sound treatments (control or noise); if selected for the noise treatment, it was also randomly assigned audio 1 or 2. Separate crabs were used for antipredator and feeding disruption experiments to remove compounding effects from repetitive handling stress or noise exposure. Ninety-six crabs (16 individuals from each site) were tested in the antipredator experiment (N = 96) and the feeding disruption experiment (N = 96), with eight individuals per sound treatment group (n = 8).

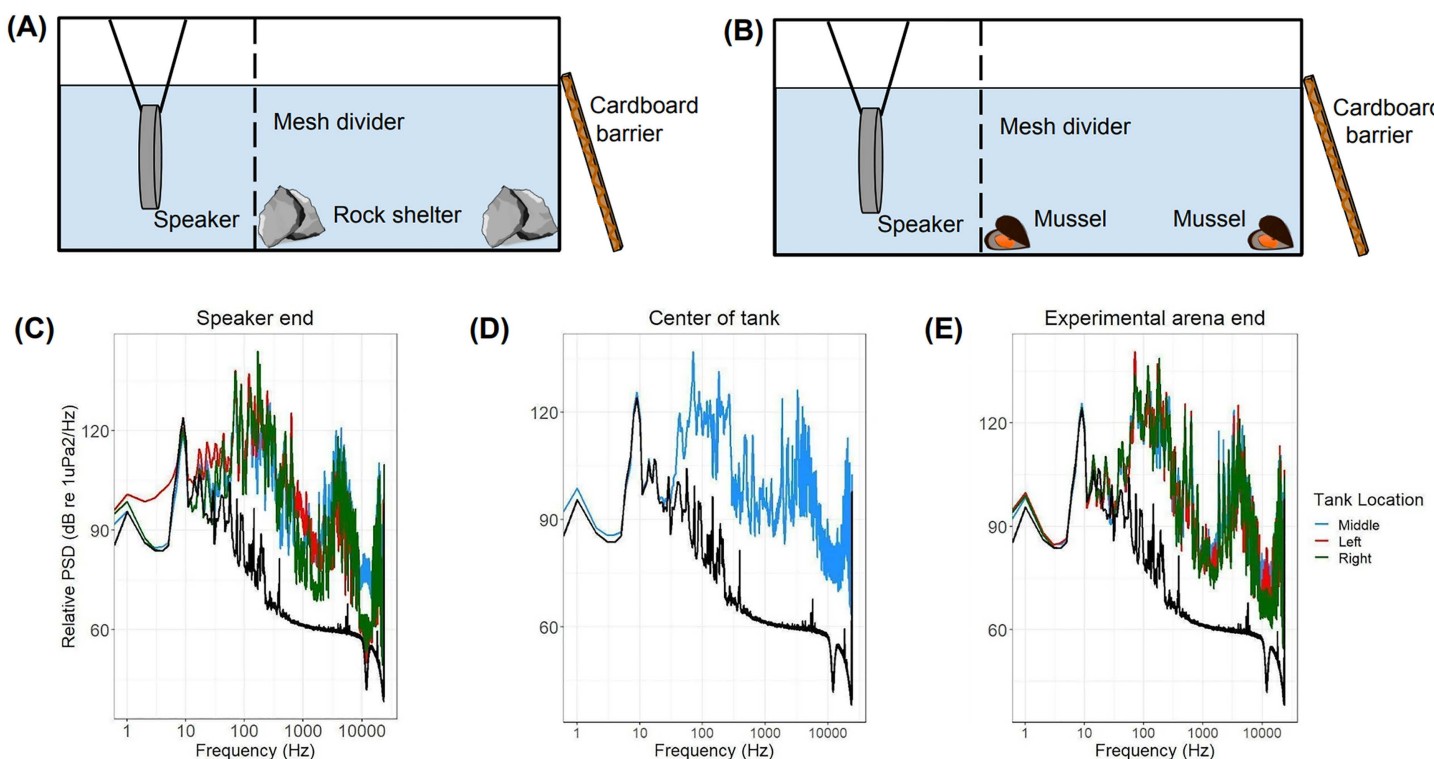

**Fig 2. Experimental tank design and analysis of sound levels. (A)** Schematic representation of tank design for antipredator trials. **(B)** Schematic representation of tank design for feeding disruption experiments. **(C, D, E)** Relative power spectral density (PSD) (dB re 1uPa$^2$/Hz) across the 20-24000 Hz frequency band for the first minute of the audio playback, measured at respective locations within the experimental tank. Colored lines denote PSD levels during noise track playback. Black lines indicate PSD levels of ambient sounds.

## Antipredator experiments

Two shelters were created in each corner of the experimental arena using two large rocks stacked together (Fig 2A). An individual *H. oregonensis* was moved from the holding tank and placed in a clear plastic tube in the middle of the two shelters, preventing the crab from retreating to shelter before the experiment began (adapted from Carter et al. [10]). The audio recording was then initiated, and the crabs were given one minute to acclimate. The plastic tube was removed, and a wooden dowel was plunged down directly in front of the crab (within 1 cm) three times to simulate a predator attack from a shore bird, a common predator of *H. oregonensis* [10,21]. The initial response of the crab was categorized as freezing or movement. The time taken for the crab to retreat to shelter was recorded (including their initial response) using a stopwatch. Crabs were considered to have retreated to shelter if their entire body was hidden by the rocks for at least 3 seconds. If the crabs did not retreat to shelter after one minute, the trial ended, and these individuals were assigned a value of 60 seconds for the time taken to retreat (as per Carter et al. [10]).

## Feeding disruption experiments

Prior to feeding disruption experiments, food was withheld for seven days. *H. oregonensis* hunger significantly heightens at 22 days of food deprivation [29], meaning seven days causes crabs to be hungry but not starving. Four mussels (*Mytilus trossulus*) were cracked open and placed throughout the experimental tank (Fig 2B). An individual *H. oregonensis* was moved from the holding tank to the experimental tank. The crab was then allowed to find the food source without any noise playback. The selected audio recording was initiated once the crab began eating (as per Wale et al. [21]). Feeding was considered disrupted if the crab moved away from the food or exhibited a complete cessation of movement. The trial ended after 1 minute of noise (as per Wale et al. [21]).

## Literature analysis

Published literature was examined to determine the extent to which this study's findings align with broader research on the topic. Relevant studies were identified using a combination of search engines (i.e., Web of Science) and terms. This process identified 17 studies that met the following criteria: original research on marine arthropods, behavioral responses to sound, an experimental control, and reported sample sizes and means with either standard deviation or standard error. This process emulated previous meta-analyses of this topic [7,8]. However, the methodology was less rigorous as it lacked formal search terms and a systemic review. The mean, variance (standard deviation or standard error), and sample size were extracted for the treatment and control groups of each relevant comparison within a study, with multiple comparisons commonly occurring within each study. A total of 71 comparisons (i.e., data points) were extracted from the 17 studies.

The data from this study's antipredator experiments were extracted following the same process as the published literature. The high and low noise sites were merged, as these experiments determined that previous exposure does not influence behavior (see Results). For each metric (i.e., initial response, sheltering response), the treatment group consisted of crabs from each site exposed to noise, and the control group consisted of crabs from each site not exposed to noise. For both treatments, data were pooled by site, and site was the level of replication (n = 6). This process created a mean, variance, and sample size for the treatment and control groups for the initial response and sheltering response metrics, mirroring the data extracted from the literature. Therefore, this meta-analysis included 73 data points from 18 studies (Supplementary Material S4 Table in S1 File).

## Data analysis

Statistical analyses were performed using R version 4.3.2 [30]. Prior to data analysis, a power analysis confirmed that the sample size provided adequate statistical power (≥0.8). Due to a non-normal distribution, the time taken to retreat to

shelter was transformed into a binary variable (retreated to shelter vs. did not retreat to shelter). Using the lme4 package (v 1.1−34), we fit a generalized linear mixed effects model using a binomial distribution. Treatment, the noise level of sites, and the interaction between these two explanatory variables were included as fixed effects, as well as site nested within noise level as a random effect. We then tested the fit of this model with a type III sum of squares ANOVA using the car package (v 3.1−2). Similarly, the initial response to a simulated predator attack was analyzed in the same fashion. Model diagnostic plots were made to ensure the homoscedasticity of residuals. A post-hoc marginal means test was carried out on the initial response data to determine the effect size of the treatment. Statistical analyses were not conducted to examine the effects of noise treatment on feeding disruption, as 100% of the crabs continued feeding in both the noise treatment and the control.

The 'Metafor' package was used to calculate the effect size (Hedge's g) and the corresponding variance of each comparison [31]. The effect sizes and variances for the initial and sheltering responses were plotted. These values illustrated the ratio of the crabs that engaged in the response relative to those that did not. A zero line was plotted to indicate no difference (i.e., variances overlap the line), with values above the line signifying an increase in the ratio and below the line the reverse [7–8]. A forest plot modeled the summary effect (i.e., the weighted average of each study's effect sizes) and confidence intervals using a restricted maximum-likelihood estimator [31]. Again, a zero line was added to improve visual discernment of the response's magnitude and directionality. The studies in the forest plot were ordered by increasing value to illustrate the current research's position relative to previously published literature. A meta-analytic multivariate mixed-effect model was used to examine the relationship between response categories and the type of sound. This model was fit with the response and sound type as fixed effects and data point ID nested within the study as a random effect. A funnel plot examined the heterogeneity of the data to identify potential publication biases and model residuals were examined for normality using studentized quantile-quantile (Q-Q) plots (Fig S2 in S1 File). Model outputs were plotted to visualize the relationship between response categories and the type of sound.

## Results

### Antipredator experiments

*H. oregonensis* showed a significant difference in the initial response after a simulated predator attack between treatment groups (Type-III Wald $\chi^2$, $\chi^2 = 5.18$, $p = 0.0229$) but not between low noise versus high noise collection sites (Type-III Wald $\chi^2$, $\chi^2 = 0.0002$, $p = 0.990$), and there was no interaction between treatment and collection site noise level (Type-III Wald $\chi^2$, $\chi^2 = 0.0549$, $p = 0.815$); Fig 3A. A post-hoc marginal means test to determine effect size of the treatment demonstrated the probability that crabs in the noise treatment moved was 0.66 (95% CI: 0.49, 0.78) versus 0.32 (95% CI: 0.20, 0.49) in the control treatment.

The proportion of *H. oregonensis* that retreated to shelter after a simulated predator attack was not significantly influenced by treatment (Type-III Wald $\chi^2$, $\chi^2 = 0.000$, $p = 1.00$), collection site type (Type-III Wald $\chi^2$, $\chi^2 = 1.12$, $p = 0.291$), or their interaction (Type-III Wald $\chi^2$, $\chi^2 = 0.668$, $p = 0.414$); Fig 3B and Supplementary Material S3 Table in S1 File. See Supplementary Material S5 Table in S1 File for the mean and standard error of time to retreat to shelter after a simulated predator attack and Fig S1 in S1 File for a graphical representation of these data.

### Feeding disruption experiments

The feeding disruption experiments demonstrated uniform feeding behaviors across all 96 crabs included in the study. All individuals continued feeding for the full one minute of the trial. There were no observations of crabs ceasing movement or moving away from the food. Consequently, statistical analyses were not performed, reflecting the absence of variance in feeding disruption responses.

## Literature analysis

We incorporated our results into a broader meta-analysis of arthropod responses to noise in an attempt to identify any emerging generalities. To enable comparisons between our data and published studies, we converted the initial and sheltering responses to ratios (see Methods), allowing for effect sizes to be determined. The results of this approach matched the findings of the binary analysis (Fig 3); noise exposure increased the rate at which individuals moved but did not significantly reduce the likelihood they took shelter (Fig 4A). The forest plot (Fig 4B) illustrates that the magnitude of these responses fell within the distribution of published data, with most literature research having more negative effect sizes. However, the overall effect size (ES) of −0.059, with standard error estimates of −0.121 and 0.002, was not significant (Fig 4B). Behavioral responses to sound varied and significant effects were only seen for a subset of responses to anthropogenic noise (Fig 4C). Arthropods exposed to anthropogenic noise were more likely to engage in anti-predator responses, such as movement considered in this study (ES 0.667, confidence intervals (CI): 0.326, 1.007). However, rates of shelter-seeking and camouflage decreased when exposed to anthropogenic noise (ES 0.122, CI: −0.645, −0.164). All other responses overlapped the zero line, signifying non-significance.

## Discussion

In this study, we analyzed the effects of shipping noise on the antipredator response and feeding behavior of the shore crab *Hemigrapsus oregonensis*, and examined the potential for these crabs to develop tolerance to such noise. Our findings reveal that shipping noise significantly impacted the shore crabs' initial response following a simulated predator attack, with a 66% likelihood of movement in noise-exposed individuals compared to 32% in controls across all collection sites. However, this response metric provided no evidence for the development of tolerance, as crabs collected at high noise sites performed similarly to those collected at low noise sites and did not show reduced sensitivity to noise. Further, neither experimental exposure to shipping noise nor the collection site significantly impacted whether the crabs retreated

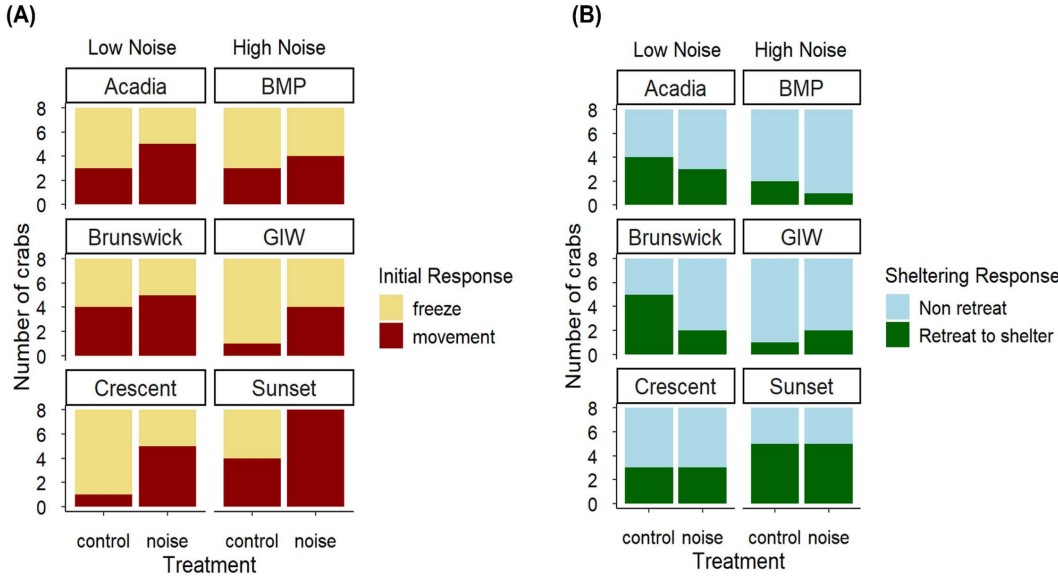

**Fig 3. Responses of *H. oregonensis* to simulated predator attacks. (A)** Initial response of *H. oregonensis* after a simulated predator attack for each treatment (noise or control) for low noise and high noise sites of origin (n = 8 per treatment per site; Treatment groups: $p = 0.0229$; Site: $p = 0.990$; Treatment and site interaction: $p = 0.815$). **(B)** Sheltering response of *H. oregonensis* after a simulated predator attack for each treatment across sites (n = 8 per treatment per site; Treatment groups: $p = 1.000$; Site: $p = 0.291$; Treatment x site interaction: $p = 0.414$). The feeding disruption experiments (not shown here) demonstrated a uniform response across all sites.

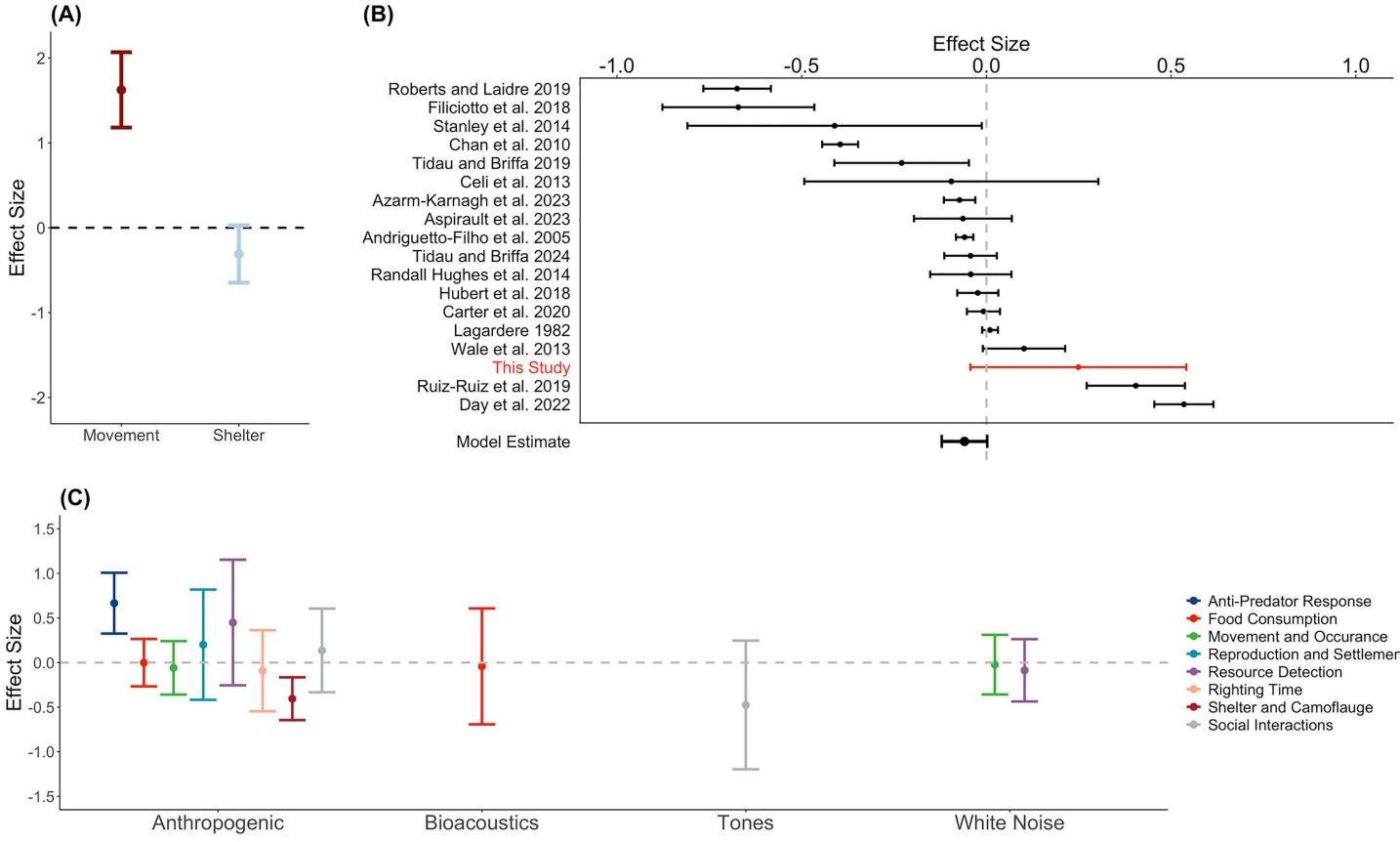

**Fig 4. Comparison of the current study and published literature. (A)** Effect sizes and variance from the current study's initial response (movement) and sheltering response (shelter). **(B)** Forest plot of published research and the current study, ordered by effect size value. Bars denote corresponding sample variance. **(C)** Model estimates and 95% confidence intervals for behavioral responses to each of the sound types. For all panels, the dashed line indicates an effect size of zero, and non-overlapping 95% CIs are significant.

to shelter or ceased feeding. Our findings align with the broader literature on how marine noise influences arthropod behavior as responses can be positive, negative, or null.

Our observation that shipping noise significantly influenced the initial response of *H. oregonensis* to a simulated predator attack warrants careful interpretation. Freezing or moving from a predator are typical behaviors observed in crabs [21]. However, moving can be a costly behavior if the movement does not result in obtaining shelter, as the crab would remain susceptible to attack, and movement reduces a crab's likelihood of blending into the background. In our study, the majority (18/31) of noise-exposed crabs that initially moved did not subsequently retreat to shelter, indicating that 58% of movements were non-directional. In contrast to our findings, others found no significant difference in the initial response of crabs to a simulated predator attack [21]. These differences could be attributed to noise being a distraction that decreases attention to tasks such as foraging [32] and predator detection [33,34]. The observed non-directional movement might demonstrate the "distracted prey hypothesis", wherein a portion of the crab's finite attention is reallocated away from the primary task [33]. However, the absence of a significant difference between our treatment groups in terms of whether they retreated to shelter does not support this hypothesis. Conversely, the non-directional movement could be attributed to the crabs' orientation during the simulated predator attack, possibly not being ideally positioned to perceive the shelter. Our experimental design aimed to mitigate this variable but due to the large tank size it is still possible this limited the crabs' ability to detect the presence of shelter, both with and without the playback.

The results of our study show that *H. oregonensis* was not impacted by shipping noise during feeding, as no individuals were observed to be disrupted. Our results are consistent with some previous research, which found that shipping noise had no significant impact on the foraging duration of *Carcinus maenas* [21,22]. However, shipping noise (148–155 dB re 1 μPa) has been shown to disrupt the feeding activities of *C. maenas*, making them more likely to suspend feeding [21]. Additionally, an in-situ study found *C. maenas* had reduced aggregation around a food item in the presence of white noise (129.5–142 dB re 1 μPa) [35]. Hubert et al. [22] proposed that the difference in results between the studies could be attributed to the variation in hunger. In our study, the crabs were deprived of food for seven days prior to the trials, whereas Wale et al. [21] deprived crabs for three days before beginning their trials, and Hubert et al. [35] were unable to quantify the hunger levels of the crabs because they performed their study in situ. The heightened hunger levels in our study may have increased the crabs' motivation to continue feeding despite potential disturbances. This postulation aligns with findings that crabs' behaviour is hunger dependent, as food selectivity being negatively correlated with starvation has been documented in *Hemigrapsus sanguineus* and *C. maenas* [36–37]. The absence of an impact of shipping noise on feeding behavior in our study could be attributed to these elevated hunger levels, which may have made the crabs more determined to feed. Future research should examine the effects of anthropogenic noise on crabs subjected to a shorter period of food deprivation to understand this relationship better.

Our study demonstrated no significant difference in whether crabs retreated to shelter after a simulated predator attack. These findings align with Wale et al. [21], who found that shipping noise did not affect the probability of crabs responding to a simulated predator attack. Conversely, others reported that shipping noise significantly reduced the probability of crabs retreating to shelter following such an attack [10]. The differences in these results may be attributed to the different life stages used in the experiments, as Carter et al. [10] used juvenile crabs, whereas both our study and Wale et al. [21] used adult crabs. It has been proposed that juvenile crabs may be more susceptible to the "distracted prey hypothesis" [33], possibly due to their immaturity, which could increase their response to environmental stimuli such as noise [38]. Furthermore, other environmental pollutants have been shown to significantly impact juvenile crabs while having no effect on adults [39]. Juveniles are inherently more susceptible to predation compared to adult *C. maenas* [40], so this vulnerability may contribute to the observed differences in their antipredator responses.

The difference in antipredator responses and feeding disruption between our findings and previous research may indicate a species-specific response of *H. oregonensis* to shipping noise. Prior studies have shown species-specific responses to noise pollution among fish, attributed to variations in life history traits and physiological stress reactions [41–43]. For example, minnows reduced foraging behavior in the presence of shipping noise, whereas sticklebacks maintained their foraging effort but made more mistakes while doing so [41]. Similarly, differential responses to noise have been observed across invertebrate species [9], suggesting that *H. oregonensis* might possess distinct life history characteristics or adaptations that modulate its reaction to noise pollution in contrast to *C. maenas.* In British Columbia, Canada, *C. maenas* is an invasive species [44] and is a predator to *H. oregonensis* [45]. Therefore, future studies are needed to investigate differing life histories or adaptations within *H. oregonensis* and *C. maenas* to better understand their responses to noise, as these may have important future consequences.

We found that the interaction between treatment and collection site was not significant for any of the behavioral metrics measured, suggesting that *H. oregonensis* do not develop tolerance to shipping noise, at least for our response metrics. Wale et al. [19], suggested that the behavioral changes in *C. maenas* observed could be attributed to the crabs developing tolerance to repeated exposure to shipping noise. However, another study by Carter [38] found no evidence of juvenile *C. maenas* developing tolerance to shipping noise. Carter [38] proposed that the variability and intermittency associated with ship passings could be hindering the development of tolerance, especially as their study exposed the crabs to the noise for only eight weeks. However, this explanation is unlikely to apply to the current study, as the crabs from the high noise sites have been exposed consistently to a variety of shipping noises. For example, crabs from Girl in a Wetsuit Beach experience significant and varied shipping traffic given its location at the entrance to various ports and marinas,

including the Port of Vancouver, which received over 2,000 commercial vessels in 2022 [46]. However, vessels are not always present through the day or year, for example, larger numbers of smaller vessels are observed during the day, on weekends, and summer months [47], leading to variability in vessel noise which could be attributing to reduced tolerance. Additionally, playback sound levels were over 10 dB re 1 µPa higher compared to the loudest site (Girl in a Wetsuit). This difference in sound levels could cause all crabs to respond regardless of site of origin, removing the ability to detect tolerance in the current study. Future work to understand tolerance should record multiple days of soundscape data from collection sites to understand the range and temporal patterns in sound levels.

The combination of no evidence for tolerance and the lack of significant impact on sheltering and feeding behavior suggests that noise is likely a pollutant that causes nuanced sublethal impacts on *H. oregonensis*. While noise did influence the initial response of *H. oregonensis* following a simulated predator attack, its actual impact on the crabs' survival and fitness, if any, remains uncertain. This finding highlights the importance of examining marine invertebrate responses to noise pollution instead of drawing conclusions from studies on fish or marine mammals [7,9]. Sound is vibrational energy with particle motion and pressure components, and as such, species' responses to a sound depend on how the acoustic characteristics of the sound are perceived [42,48]. As crustaceans and other invertebrates primarily detect particle motion components of sound, noise may not elicit responses observed in higher-order taxa [12,48]. In some instances, noise levels that cause stress responses in marine fish or mammals may act as a settling cue for marine invertebrates [12,15]. Our examination of the broader literature affirms this consideration, as responses to noise that are well established within the study of some vertebrates, such impacts on reproduction [7], were not observed.

To gain insight into the potential reasons behind these results, analyzing the sound levels within the tank offers additional context. This analysis revealed that the highest sound levels (~125–140 dB re 1 µPa) of the noise track playback occurred within the frequency range of 65–275 Hz (Fig 2C, 2D, 2E). Invertebrates are most sensitive to low-frequency noise, typically ranging from 300 to 2000 Hz [38]. The noise tracks were recordings of three different types of ships (a tanker, a ferry, and a recreational fishing boat) collected from the same sites as the crabs. Therefore, the shipping noise playback may not have overlapped the frequency range that *H. oregonensis* are most sensitive to. Consequently, the lack of sheltering and disrupted feeding could be attributed to a lack of sensitivity, although the significant initial response suggests that the crabs were able to perceive these sounds. Conducting specific sensitivity tests tailored to *H. oregonensis* would provide insights into their precise sensitivity range and help elucidate their response to noise pollution more accurately.

Extrapolating behavioral findings from controlled tank studies to animals in their natural environments requires careful consideration [49], particularly with acoustic playback experiments. Playback experiments struggle to replicate natural sound environments because the complex sound field within a tank does not allow sound to propagate as it would in an open environment [50,51]. Additionally, crabs are more sensitive to the particle motion component of sound rather than the pressure component [13], the former of which was not measured due to technical constraints [26]. While our study may not provide precise sensitivity measurements, it does offer valuable insights into behaviors potentially affected (or unaffected) by the introduction of shipping noise in controlled, repeatable, and comparable conditions. However, evidence suggests sound pressure and particle components of acoustic recordings can be retained during playback [52], and particle acceleration correlates with sound pressure levels [48]. To refine our understanding further, future research should incorporate accelerometers into laboratory studies to measure particle motion as current insight are derived from in situ measurements [52,48].

*Hemigrapsus oregonensis* are a highly abundant species within the intertidal ecosystems of the Northeast Pacific Ocean [53], underscoring the importance of investigating the potential impacts of shipping noise on their behavior. Our results indicate that *H. oregonensis* may not develop tolerance to shipping noise, potentially due to its limited impact as a biological pollutant for these shore crabs, as evidenced by its non-significant effects on two of the three behavioral responses examined in this study (sheltering and feeding). These findings are somewhat counterintuitive given documented impacts of noise on marine mammals and fish [7,9]. However, a meta-analysis of related data indicated our

findings are in keeping with the broader literature, as sound can have a positive, negative, or null effect on marine arthropods. Such findings are crucial as they enhance our understanding of which organisms are susceptible to the adverse effects of shipping noise and which are less impacted. Knowledge of species-specific impacts equips us to better inform conservation efforts and formulate effective marine noise pollution policies.

## Supporting Information

**S1 File.** **S1 Table.** Relative Sound Pressure Level (SPL) Range at Selected Sites. **S2 Table.** Temperature and Salinity Measurements at Selected Sites. **S3 Table**. Proportion of Individual *H. oregonensis* That Retreated to Shelter After a Simulated Predator Attack. **S4 Table.** Effective sizes and variance for 71 data points from 17 studies that examined sound influences on marine arthropods' behavior. Estimates for the current study are also included here. See the supplemental reference list below for additional information on the studies. **Table S5.** Mean and Standard Error of Time to Retreat to Shelter After a Simulated Predator Attack. **Fig S1.** Sheltering Response Time of *H. oregonensis* to Simulated Predator Attacks. Time taken (s) for *H.oregonensis* to retreat to shelter after a simulated predator attack for each treatment according to site. n = 8 per treatment per site. **Fig S2. A)** A funnel plot examining the heterogeneity of the data to identify potential publication biases. B) Studentized quantile-quantile (Q-Q) plot of model residuals used to examine model fit and normality of the data.
(DOCX)

## Acknowledgments

We thank Francis Juanes for providing the equipment necessary to complete this project. We also thank Anika Chen and Cassidy Mark for their assistance with fieldwork. This study was conducted on the ancestral and unceded territories of the xʷməθkwəy̓əm (Musqueam), Skwxwú7mesh (Squamish), Səl̓ílwətaʔ/Selilwitulh (Tsleil-Waututh), sc̓əwaθən məsteyəxʷ (Tsawwassen), Stó:lō, Semiahmoo, and q̓ic̓əy̓ (Katzie) First Nations.

## Author contributions

**Conceptualization:** Abigail Birch, Christopher D. G. Harley.

**Formal analysis:** Abigail Birch, Kieran D. Cox, Kelsie A. Murchy, Sandra Emry.

**Funding acquisition:** Kieran D. Cox, Christopher D. G. Harley.

**Investigation:** Abigail Birch.

**Methodology:** Abigail Birch, Kelsie A. Murchy, Christopher D. G. Harley.

**Project administration:** Abigail Birch.

**Resources:** Kelsie A. Murchy, Christopher D. G. Harley.

**Supervision:** Kieran D. Cox, Christopher D. G. Harley.

**Validation:** Abigail Birch, Kieran D. Cox, Kelsie A. Murchy, Sandra Emry, Christopher D. G. Harley.

**Visualization:** Abigail Birch, Kieran D. Cox, Kelsie A. Murchy.

**Writing – original draft:** Abigail Birch.

**Writing – review & editing:** Abigail Birch, Kieran D. Cox, Kelsie A. Murchy, Sandra Emry, Christopher D. G. Harley.

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
