## [Decision Letter · Decision Letter 0]

28 Apr 2025

PONE-D-25-08170Shipping noise tolerance in the shore crab Hemigrapsus oregonensisPLOS ONE

Dear Dr. Harley,

Thank you for submitting your manuscript to PLOS ONE. After careful consideration, we feel that it has merit but does not fully meet PLOS ONE’s publication criteria as it currently stands. Therefore, we invite you to submit a revised version of the manuscript that addresses the points raised during the review process.

Unfortunately we were only able to find one reviewer but that reviewer was generally supportive with only minor comments that need addressing. Please ensure that your decision is justified on PLOS ONE’s publication criteria  and not, for example, on novelty or perceived impact.

We look forward to receiving your revised manuscript.

Kind regards,

Judi Hewitt

Academic Editor

PLOS ONE

 [Funding was provided by an NSERC Discovery Grant (RGPIN-2022-04683) to CDGH and the Liber Ero Fellowship Program to KDC. AB was supported by an Ecosystems and Oceans Science Contribution Framework grant awarded to KDC and Francis Juanes.]. 

4. In the online submission form, you indicated that [Data will be submitted to a public repository upon manuscript acceptance. We are happy to provide the data to the reviewers if requested.].

5. In the online submission form, you indicated that your data will be submitted to a repository upon acceptance.  We strongly recommend all authors deposit their data before acceptance, as the process can be lengthy and hold up publication timelines. Please note that, though access restrictions are acceptable now, your entire minimal  dataset will need to be made freely accessible if your manuscript is accepted for publication. This policy applies to all data except where public deposition would breach compliance with the protocol approved by your research ethics board. If you are unable to adhere to our open data policy, please kindly revise your statement to explain your reasoning and we will seek the editor's input on an exemption.

6. PLOS requires an ORCID iD for the corresponding author in Editorial Manager on papers submitted after December 6th, 2016. Please ensure that you have an ORCID iD and that it is validated in Editorial Manager. To do this, go to ‘Update my Information’ (in the upper left-hand corner of the main menu), and click on the Fetch/Validate link next to the ORCID field. This will take you to the ORCID site and allow you to create a new iD or authenticate a pre-existing iD in Editorial Manager.

7. Your ethics statement should only appear in the Methods section of your manuscript. If your ethics statement is written in any section besides the Methods, please move it to the Methods section and delete it from any other section. Please ensure that your ethics statement is included in your manuscript, as the ethics statement entered into the online submission form will not be published alongside your manuscript.

8. We note that Figure 1B in your submission contain [map/satellite] images which may be copyrighted. All PLOS content is published under the Creative Commons Attribution License (CC BY 4.0), which means that the manuscript, images, and Supporting Information files will be freely available online, and any third party is permitted to access, download, copy, distribute, and use these materials in any way, even commercially, with proper attribution. For these reasons, we cannot publish previously copyrighted maps or satellite images created using proprietary data, such as Google software (Google Maps, Street View, and Earth). For more information, see our copyright guidelines: http://journals.plos.org/plosone/s/licenses-and-copyright.

1. You may seek permission from the original copyright holder of Figure 1B to publish the content specifically under the CC BY 4.0 license. 

Additional Editor Comments (if provided):

Reviewers' comments:

Reviewer's Responses to Questions

**Comments to the Author**

1. Is the manuscript technically sound, and do the data support the conclusions?

Reviewer #1: Yes

2. Has the statistical analysis been performed appropriately and rigorously? 

Reviewer #1: Yes

3. Have the authors made all data underlying the findings in their manuscript fully available?

Reviewer #1: Yes

4. Is the manuscript presented in an intelligible fashion and written in standard English?

Reviewer #1: Yes

5. Review Comments to the Author

Reviewer #1: The menauscript is truly well done and, despite focused on a species, support reflections on the general approach to consider the impacts of shipping noise on ecosystems' health. Additional comments are uploaded for minor changes.

6. PLOS authors have the option to publish the peer review history of their article (what does this mean? ). If published, this will include your full peer review and any attached files.

**Do you want your identity to be public for this peer review?** For information about this choice, including consent withdrawal, please see our Privacy Policy .

Reviewer #1: **Yes: ** Pier Francesco Moretti

---

## [Author Response · Author response to Decision Letter 1]

8 Jul 2025

[This information is also included in the cover letter.]

Reviewer 1 Comments to the Author:

General Comment:

The manuscript presents results of original research, and not been published elsewhere, and written in standard English.

Experiments, statistics, and other analyses are performed to a good technical standard and are described in sufficient detail.

The research meets all applicable standards for the ethics of experimentation and research integrity.

Conclusions are presented in an appropriate fashion, supported by the data. Despite focused on a single species, it provide argumentation for reflections and further developments.

Author Response

We thank Reviewer 1 for their positive assessment of our manuscript and for recognizing the quality of our experimental design, statistical analysis, and conclusions. We appreciate the encouraging comments regarding the integrity of the research, and we are glad that the manuscript’s broader implications were clear despite the species-specific focus.

Requests for minor revision

Reviewer Comment

Title: the title would be more catching if includes a more general aspect: “Shipping noise tolerance in invertebrates: the case of shore crab Hemigrapsus oregonensis”

Author Response

We appreciate the reviewer’s suggestion to broaden the title’s appeal. In response, we have revised the title to highlight the relevance of our findings to invertebrates more generally. The new title reads: “Shipping noise tolerance in invertebrates: A case study of the shore crab Hemigrapsus oregonensis”

Reviewer Comment

Row 141: what Zoom amplifier has been used? Probably the F6? At what sampling frequency? Please clarify.

Author Response

We used a zoom recorder, H1 handy recorder to save the wav files to an SD card. We added

clarifying details. Line 143-144: “…connected to a ZOOM recorder (H1 Handy Recorder, sampling rate = 48 kHz, 16-bit)…”

Reviewer Comment

Row 158: when the 1h recordings were obtained? There should be a difference between day and night, especially when linked to living organisms’ circadian rhythms. Please clarify and add a comment on this aspect.

Author Response

We thank the reviewer for pointing this out. We have clarified the timing of the recordings in the Methods section on Row 140 which now reads: “At each site, a one-hour recording was made between 09:30 and 17:00 using an underwater hydrophone (Cetacean Research Technology SQ-26; sensitivity: 169 dB re 1 V/µPa; frequency range: 0.020-50 kHz) connected to a ZOOM recorder (H1 Handy Recorder) at one meter depth.” We also revised Row 158 to: “Fig 1. Acoustic Analysis of Study Sites in the Lower Mainland, BC. (A) Relative sound pressure levels at selected sites based on one-hour underwater recordings made between 09:30 and 17:00 using a SQ-26 hydrophone and ZOOM recorder.”

Reviewer Comment

Fig.1 and caption: the figure shows 6 sites, even you implemented the experiments in 9 sites. You missed additional 3 high noise sites in the map (probably because far from the others?). In case, include in the caption the 3 high noise sites that are not displayed.

Author Response

We thank the reviewer for this comment. To clarify, although nine sites were initially surveyed to assess ambient noise levels, only six were ultimately included in the experimental component of the study, based on their relative SPL values and the presence of suitable crab populations. We have revised the first sentence of the Methods section to make the site selection process clearer. As the final study was implemented at only six sites, we have chosen to display only these in Figure 1.

Reviewer Comment

Fig.3 and caption: Specify that the initial response is to the simulated predator attack. Add in the caption that the feeding experiments gave 100% uniform response (this is the reason why you do not show in the figure). Again you show 6 sites instead of 9? What am I missing?

Author Response

We have now included a brief mention of the uniform feeding disruption response in the caption, which was why the data are not shown in the figure. The initial response to the simulated predator attack is already specified in the figure caption. Regarding the number of sites, the experiment was only implemented at six sites, not nine. We have clarified this in the Methods section to avoid any confusion.

Reviewer Comment

Main Issue 1: you refer to SPL, but you do not report any information about calibration of the hydrophone+zoom. Please include how calibration has been implemented and the accuracy of sensitivity across frequencies.

Author Response

We used the hydrophone sensitivity provided by the manufacturer, but did not calibrate prior to

deployment. This was consistent to what we found in the literature for these hydrophones, see Lillis

et al. 2018 Drifting hydrophones as an ecologically meaningful approach to underwater soundscape measurement in coastal benthic habitats, so we feel that this is justified. We changed the language to relative SPL, to account for the variation that could be present. Note that we have recently purchased a pistonphone that could be used to calibrate post-hoc if required. However, we added more details on the hydrophone, including the frequency accuracy across frequencies. Lines 141-143: “…underwater hydrophone (Cetacean Research Technology SQ-26; sensitivity: -

169 dB re 1 V/µPa; flat frequency response (± 1 dB) up to 28 kHz, frequency range: 0.020-50

kHz)”

Reviewer Comment

Main issue 2: please clarify why you choose the bird as a predator and provide a spectrum of the signal you introduced to simulate the predator attack. Samples of the spectra of the shipping noise in the 9 sites would also be welcome in the supporting information.

Author Response

Thank you for the comment. We have added a point to Row 223 to clarify that the shore bird was chosen because it is a common predator of H. oregonensis. To clarify, the predator attack itself did not involve noise; rather, shipping noise was played in the background to test its effect on the crab’s behavior, specifically its time to retreat to shelter. The spectrum of the shipping noise used in the experiment can be seen in Figure 2.

Regarding the sound data, we only included the SPL data for the six sites selected for the study, as the other three sites either had incomplete SPL data or did not have crabs, making them unsuitable for inclusion. This information is provided in Table S1 of the supplementary materials.

---

## [Editor Report · Decision Letter 1]

11 Jul 2025

Shipping noise tolerance in invertebrates: A case study of the shore crab Hemigrapsus oregonensis

PONE-D-25-08170R1

Dear Dr. Harley,

We’re pleased to inform you that your manuscript has been judged scientifically suitable for publication and will be formally accepted for publication once it meets all outstanding technical requirements.

Kind regards,

Judi Hewitt

Academic Editor

PLOS ONE
---

## [Editor Report · Acceptance letter]

PONE-D-25-08170R1

PLOS ONE

Dear Dr. Harley,

I'm pleased to inform you that your manuscript has been deemed suitable for publication in PLOS ONE. Congratulations! Your manuscript is now being handed over to our production team.

Kind regards,

on behalf of

Dr. Judi Hewitt

Academic Editor

PLOS ONE